# Rad9/53BP1 promotes DNA repair via crossover recombination by limiting the Sgs1 and Mph1 helicases

Matteo Ferrari [1,2], Chetan C. Rawal [1,3], Samuele Lodovichi [1], Maria Y. Vietri [1] & Achille Pellicioli [1✉]

The DNA damage checkpoint (DDC) is often robustly activated during the homologous recombination (HR) repair of DNA double strand breaks (DSBs). DDC activation controls several HR repair factors by phosphorylation, preventing premature segregation of entangled chromosomes formed during HR repair. The DDC mediator 53BP1/Rad9 limits the nucleolytic processing (resection) of a DSB, controlling the formation of the 3′ single-stranded DNA (ssDNA) filament needed for recombination, from yeast to human. Here we show that Rad9 promotes stable annealing between the recombinogenic filament and the donor template in yeast, limiting strand rejection by the Sgs1 and Mph1 helicases. This regulation allows repair by long tract gene conversion, crossover recombination and break-induced replication (BIR), only after DDC activation. These findings shed light on how cells couple DDC with the choice and effectiveness of HR sub-pathways, with implications for genome instability and cancer.

[1] Dipartimento di Bioscienze, Università degli Studi di Milano, Via Celoria 26, 20131 Milano, Italia. [2] Present address: Developmental Biology Program, Memorial Sloan Kettering Cancer Center, New York, NY 10065, USA. [3] Present address: Department of Molecular and Computational Biology, University of Southern California, Los Angeles, CA 90089, USA. ✉email: achille.pellicioli@unimi.it

The first step to channel DSBs into faithful HR repair is the formation of 3′ ssDNA through resection. The nucleolytic processing is carried out through a complex and finely regulated process requiring the cooperation of several factors, including the nucleases Mre11, Exo1 and Dna2, aided by the helicase Sgs1 (BLM in human)[1]. The exposed 3′ end filament is covered by the ssDNA binding factor RPA, the recombinase Rad51 and the strand-annealing and Rad51-loading protein Rad52[2]. Rad51-nucleoprotein filament with the assistance of other proteins engages in a sophisticated and dynamic process. This involves strand invasion into the donor and annealing to a complementary sequence that serves as template for DSB repair. This process leads to the formation of a DNA joint molecule intermediate, defined as a displacement-loop (D-loop)[3]. Then, repair DNA synthesis starts from the 3′ end of the invading filament, extending the D-loop. Of note, D-loops can be extended and processed in different ways, promoting a variety of HR sub-pathways, with/without crossover (CO). Moreover, nascent D-loops can be reversed by specific helicases and/or topoisomerases (Srs2, Mph1 and Sgs1-Top3-Rmi1 (STR) in yeast; BLM-TOPIIIα-RMI1/2, FANCJ, FBH1, PARI, RECQ1, RECQ5, RTEL1, FANCM, and maybe others in human), through finely regulated mechanisms that, according to recent data in yeast, also involve the Rad54-paralog Rdh54/Tid1[4,5].

In budding yeast, the 53BP1 ortholog Rad9 acts as a scaffold, mediating signalling from the upstream DDC kinases, Mec1 and Tel1 (ATR and ATM in human), to the downstream effector kinases, Rad53 and Chk1 (CHK2 and CHK1 in human). In addition to its role in the DDC, Rad9 inhibits DSB resection[6]. Rad9 physically limits the recruitment of Sgs1-Dna2 at DSBs, while also reducing DSB repair through the Rad51-independent single-strand-annealing (SSA) pathway[7–10]. Remarkably, these functions are also conserved with 53BP1, with implication for cancer biology[11].

However, it is not understood how the formation of the 3′ filament is coupled with the selection of appropriate HR sub-pathway, and whether Rad9/53BP1 may have a role in determining that choice. Of interest, in yeast Rad9 promotes sister chromatid exchanges and limits complex chromosome rearrangements and translocations[12,13]. Moreover, Rad9 was proposed through its role in the DDC to favour BIR completion by preventing premature chromosome segregation[14,15] and also promoting the Rad53-dependent phosphorylation of the helicase Pif1. This helps nascent D-loop elongation and DNA synthesis by BIR[16].

Starting from a genetic assay and then using specialized systems to physically monitor recombination products, we find that Rad9 promotes long-tract gene conversions (GC), BIR and CO, during the HR repair of a DSB. We show that these outcomes are reduced in the absence of functional Rad9 due to the hyper-loading of the Sgs1 and Mph1 helicases. We also show that Mph1 and Sgs1 severely impaired D-loop extension in the absence of Rad9. Therefore, we propose that Rad9 restrains the recruitment of helicases involved in DSB resection and also in D-loop stability, coordinating two distinct and finely regulated steps of HR repair. This regulation couples the formation of joint DNA molecules in DSB repair with DDC activation, reducing the risk of premature and catastrophic segregation of tangled chromosomes, thus preserving genome integrity.

## Results

**Rad9 promotes long-tract GC, BIR and CO recombination**. To investigate the role of Rad9 in HR repair, we performed a DSB-induced recombination assay using a diploid yeast background (LSY2205-11C/LSY2543). This genetic background allows the study of DSB repair through GC and BIR, upon an I-SceI induced break in the ADE2 locus on chromosome XV[17]. Importantly, this system measures frequencies of non-crossover (NCO) and CO in DSB repair, and also distinguishes between short and long-tract GC (Fig. 1a; Supplementary Fig. 1). Upon I-SceI induction by the addition of galactose into the cell culture media, we observed similar plating efficiency (number of colonies in galactose/number of colonies in glucose) for wild-type and rad9Δ cells (Supplementary Table 1), allowing us to further investigate the different classes of survivors. After plating it is possible to observe three different classes of colonies in term of colour, due to recombination at ADE2 gene and segregation: white (ADE2/ade2), red (ade2/ade2) or white/red sectored colonies (ADE2/ade2 and ade2/ade2) (see schemes in Fig. 1a and Supplementary Fig. 1), depending on short-tract GC, long-tract GC and a combination of both, respectively. Moreover, each colony can be screened for specific genetic markers (HPH, NAT, MET22, TRP1), allowing the identification of different repair outcomes: CO/NCO, BIR and chromosome loss events (Fig. 1a; Supplementary Fig. 1). As an important control, all the survivors are tested by I-SceI re-induction assay that allows for the identification of the colonies (also called non-recombinants) in which I-SceI DSB was not induced in the first round of induction.

Upon I-SceI induction in the rad9Δ recombinants we observed a striking increase in the percentage of short-track GC (white colonies in the assay) (Fig. 1b, c; Supplementary Table 2). After screening each class of colonies for specific markers, we also found that rad9Δ cells have a reduced frequency of both CO and BIR events (Fig. 1c; Supplementary Table 1). Importantly, all the wild-type and rad9Δ recombinants obtained in the assay were able to grow in synthetic complete media lacking tryptophan or methionine (Supplementary Table 1), indicating that neither of them lost the cut and/or donor chromosomes. These results are in line with previous analysis done in this genetic background[17], although RAD9 deletion was shown to increase DSB-induced chromosome loss events in a different genetic system, possibly due to abortive BIR events[14].

To verify if the alteration of the HR repair might be caused by premature chromosome segregation in rad9Δ cells, we also performed the assay in cells blocked in G2/M phase by nocodazole treatment. Keeping the cells arrested in G2/M for 8 h after the DSB induction prior to plating, lead to similar results obtained in asynchronous cell populations (Fig. 1b, c; Supplementary Tables 1–4), confirming a slightly lower survival and a reduced frequency of BIR and CO events in rad9Δ cells. Based on these results, we hypothesized that Rad9 might control strand invasion-mediated mechanisms to repair a DSB, in addition to its function in limiting DSB resection and SSA.

To study more precisely the role of Rad9 in the BIR process, we took advantage of a haploid genetic system (JRL092 background) engineered to test only this specific HR sub-pathway[18]. Briefly, one DSB is induced on chromosome V by the endonuclease HO and it can be repaired by BIR with a donor sequence on chromosome XI (Fig. 2a). Analysing the DSB repair by Southern blot in exponentially growing or nocodazole-arrested cells (Fig. 2b, c), we found that RAD9 deletion severely affected BIR, regardless of the cell-cycle stage, confirming our previous observation that the BIR defect of rad9Δ cells is unlikely due to only premature chromosome segregation. We speculated the rad9Δ cells defect in BIR could result from DDC signalling to Pif1 helicase which contributes to BIR[16]. However, after plating the cells on galactose to induce HO, we observed higher cell lethality in rad9Δ cells with respect to pif1Δ and rad53-K227A chk1Δ cells, which are defective in DDC signalling downstream from Rad9 (Fig. 2d). These data support the idea that cells lacking RAD9 would be defective in BIR for an additional reason than deficient signalling to Pif1.

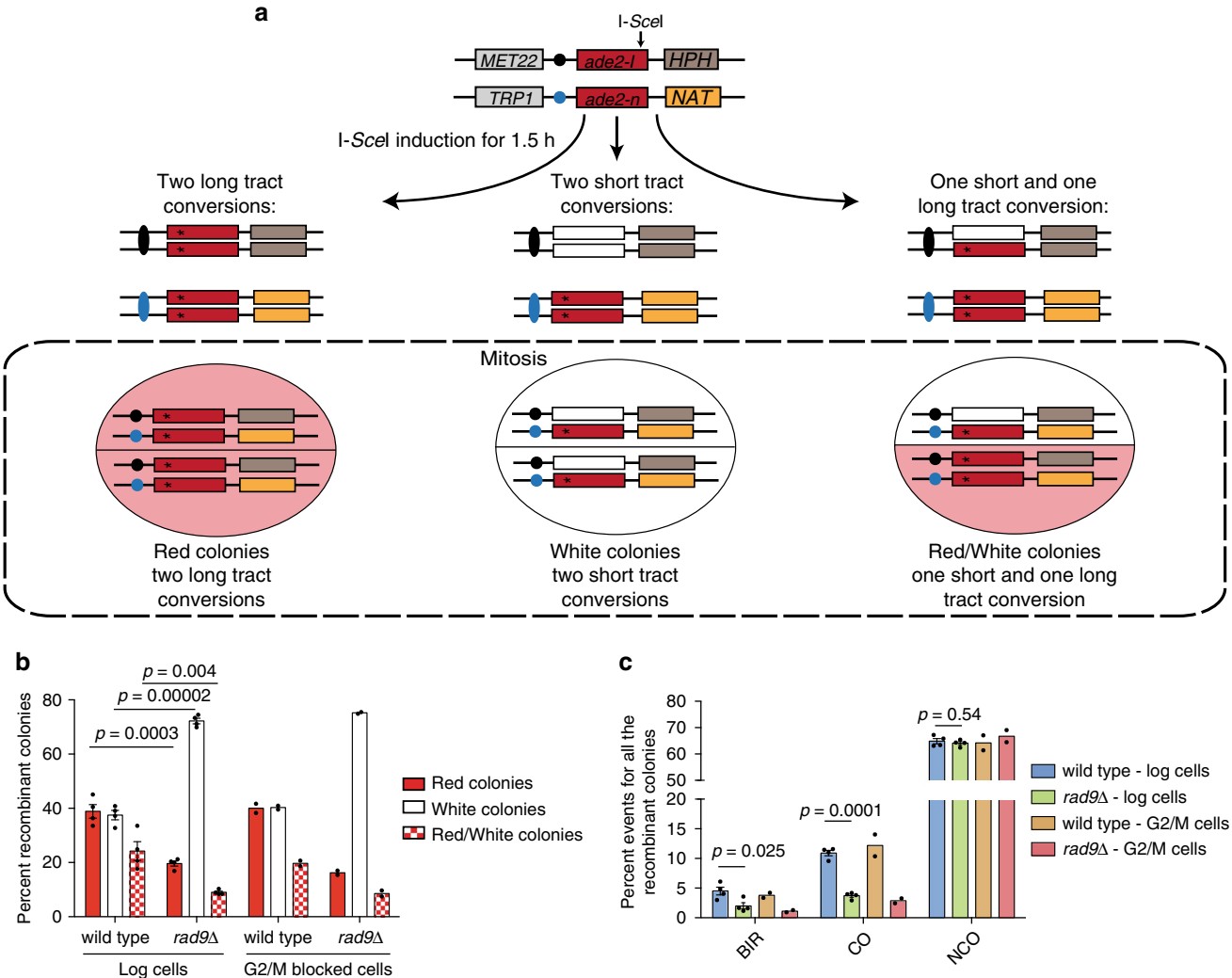

**Fig. 1 Rad9 promotes long-tract GC, BIR and CO recombination. a** Scheme of the genetic system to test recombination events in LSY2205-11C/LSY2543 background. See also Supplementary Fig. 1. **b** Distribution of the recombinant colony types in the indicated LSY2205-11C and LSY2543 derivative diploid strains. Red (two long-track conversions), white (two short-track conversions) and red/white (one long- and one short-track conversion) were determined from recombinant colonies after I-SceI induction in asynchronous ($n = 4$, genetically independent diploids) or nocodazole-arrested ($n = 2$, genetically independent diploids) cells. **c** Percent events of CO, NCO and BIR, after normalization with plating efficiency, among all the recombinant colony types of the experiments in (**b**). All the data in the figure are presented as mean ± SEM wherever $n = 3$ or more and mean where $n = 2$. Statistical analysis was done using unpaired two-tailed Student's $t$-test. See supplementary Tables 1, 3.

**Rad9 limits Rpa1, Rad52 and Rad51 hyper-loading at a DSB.** To test the possibility that Rad9 might have a role in another critical step of Rad51-dependent HR, we tested the loading of the recombination factors Rpa1, Rad52 and Rad51 at an HO-DSB by chromatin immunoprecipitation (ChIP). To avoid that the ChIP data may be influenced by different DSB repair kinetics between the wild-type and the $rad9\Delta$ cells, we performed the experiments in the donor-less JKM139 background, in which a single irreparable HO-DSB can be induced in chromosome III. Our results show that Rpa1, Rad52 and Rad51 were efficiently recruited at one irreparable HO-DSB (Fig. 3). However, their loading near the break ends was significantly higher in the absence of $RAD9$, especially at later time points. Moreover, for the same reason, we also analysed the Rad51 binding on the DSB side that is not engaged in BIR repair in the JRL092 background (Fig. 4a). Remarkably, also in this background we found higher Rad51 binding in $rad9\Delta$ than wild-type cells in the vicinity of the break (Fig. 4b). These observations exclude reduced binding of the recombination factors Rad51, Rad52 and Rpa1 at the DSB as a cause of BIR defects in cells lacking Rad9. We also speculated that the increased binding of these recombination factors proximal to the DSB could be explained by faster end resection in $rad9\Delta$ cells. To test this hypothesis, we analysed the amount of ssDNA near the DSB site and the end resection profile in wild-type and $rad9\Delta$ cells by a qPCR-based method[19]. We found that, even if $RAD9$ deletion increased the amount of resected DNA far from the DSB, the levels of the ssDNA generated very close to the break site in wild-type and $rad9\Delta$ cells were similar, in both the JKM139 and JRL092 backgrounds (Supplementary Figs. 2a–c and 3a). These data demonstrate that in wild-type cells, Rad9 does not alter the efficiency of the short-range resection but limits the long-range step of the process. By the qPCR analysis, we also verified that the total amount of the 3′ ssDNA close to the DSB end, relative to an uncut locus, was similar in wild-type and $rad9\Delta$ cells, in both the backgrounds JKM139 (Supplementary Fig. 2d) and JRL092 (Supplementary Fig. 3b). Taken together these observations with the elevated loading of the recombination factors at the DSB (Figs. 3 and 4b), we concluded that the severe

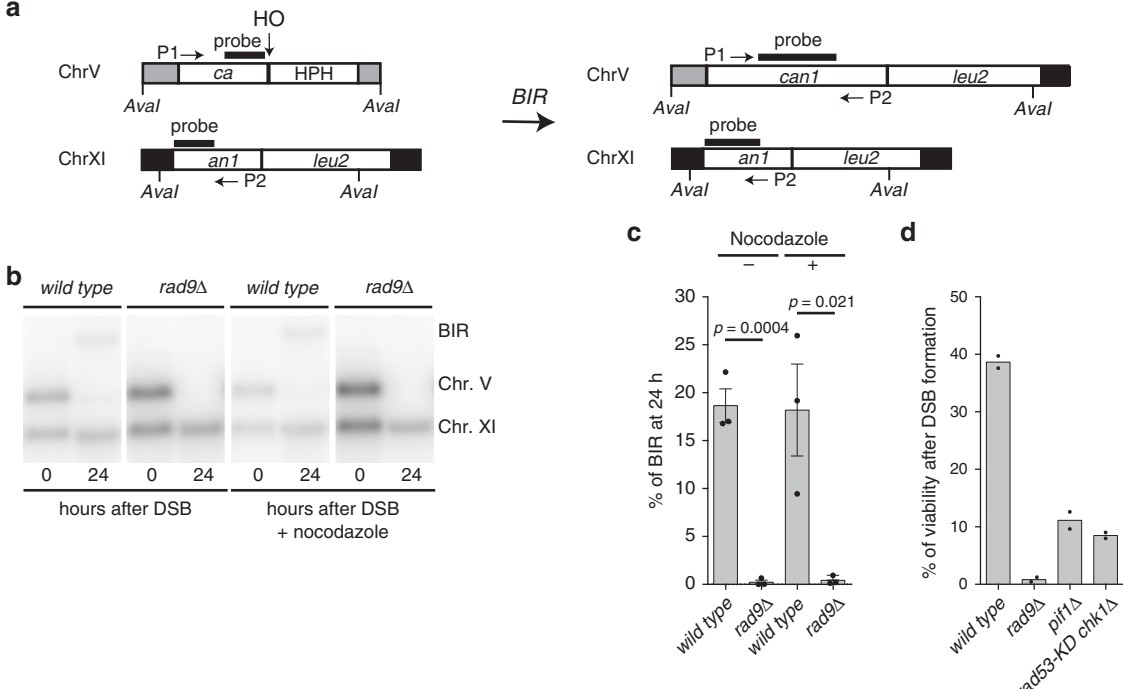

**Fig. 2 Rad9 promotes DSB repair through BIR. a** Scheme of the genetic system to test BIR in JRL092 background. **b** Southern blot of AvaI-digested DNA to monitor DSB repair through BIR in the indicated JRL092 strains. **c** Densitometric analysis of the BIR bands at 24 h of three experiments as in (**b**). Data represent mean ± SEM (n = 3, biologically independent experiments). Statistical analysis was done using unpaired two-tailed Student's t-test. **d** BIR efficiency measured by cell viability in the indicated JRL092 strains (n = 2, biologically independent experiments). See accompanying source data file.

BIR impairment of the rad9Δ cells is not due to a defect in the generation and/or stability of the 3′ ssDNA nucleoprotein filament needed for recombination. This is relevant to consider for the BIR repair in JRL092, where the relatively short homology between the cut and donor chromosomes might suggest that a possible faster degradation of the recombinogenic filament may be responsible of the severe BIR defect.

Based on the above results, it is unlikely that the increased recruitment of Rpa1, Rad52 and Rad51 close to the DSB was due to a higher amount of available substrate in rad9Δ cells. Instead, we speculated that the loading and oligomerization of Rad9 protein on the DSB might physically dampen the recruitment of the recombination factors, similar to its role in limiting nucleases for DSB resection[6–9]. To test this hypothesis, we expressed from a plasmid the wild-type Rad9 or the two protein variants Rad9-2Ala and Rad9-7xA, both reducing the Rad9 binding and oligomerization at DSBs[7,20,21], in JRL092 rad9Δ cells. We found that both the Rad9 variants, contrary to the wild-type form, did not completely rescue the lethality of rad9Δ cells in the BIR assay (Supplementary Fig. 4), supporting the idea that Rad9 might affect BIR though a physical role at the DSB site.

**Rad9 limits Sgs1 and Mph1 to promote D-loop extension.** Despite increased Rad51 binding at the DSB (Figs. 3c and 4b), we found that it was not enriched at the donor site upon DSB induction in G2/M blocked JRL092 rad9Δ cells (Fig. 4c). This result indicates that cells lacking Rad9 cannot form a stable D-loop structure and synapsis between the DSB and the donor template, providing molecular evidence of the failure in DSB repair by BIR. Consistent with these data, RAD9 deletion impaired the D-loop extension, measured through a PCR-based assay more severely than rad53-K227A chk1Δ and pif1Δ mutations (Fig. 4a, d, e). These results suggest that Rad9 promotes strand invasion and

D-loop extension in BIR, through a mechanism independent of Chk1 and Rad53 signalling. Of note, previous studies show that the annealing between the two DNA strands in D-loop formation can either be promoted by a Rad51-dependent process or rejected by the Sgs1-Top3-Rmi1 complex and Mph1[4,5,22–25]. Strikingly, the genetic deletions of the two helicases Sgs1 and Mph1 in JRL092 rad9Δ cells partially rescued cell viability (Fig. 5a), D-loop extension (Fig. 5b, c) and BIR repair (Supplementary Fig. 5), similar to the levels found in rad53-K227A chk1Δ and pif1Δ cells. Based on these data, we hypothesized that Rad9 might regulate the strand rejection during D-loop formation, limiting Sgs1 and Mph1 binding onto the recombinogenic filament. Supporting the hypothesis, ChIP of the two helicases at the DSB in JKM139 rad9Δ cells showed increased binding (Fig. 5d, e), whereas their enrichment was severely reduced at the donor site in JRL092 rad9Δ cells (Fig. 5f, g). Overall, we concluded that the binding profiles at the cut and donor sites of Sgs1, Mph1 and Rad51 were similar and might reflect the dynamic instability of the D-loop structure in rad9Δ cells.

Considering that Sgs1 physically interacts with Rpa1 and Rad51[26,27], while Mph1 binds to Rpa1 and Rad52[28,29], we speculated that the hyper-loading of both Sgs1 and Mph1 at the DSB might be related to the increased recruitment of Rad51, Rad52 and Rpa1 in rad9Δ cells (Figs. 3 and 4b). Interestingly, it was recently shown that the interaction between Sgs1 and Rad51 is abolished by the sgs1-F1192D (sgs1-FD) mutation[26]. Therefore, we tested if the sgs1-FD mutation could rescue the BIR defect of JRL092 rad9Δ cells. Indeed, we found that the sgs1-FD partially rescued the viability of cells lacking RAD9 in the BIR assay, similar to SGS1 deletion (Fig. 5h). This result suggests that a decreased interaction between Sgs1 and Rad51 might reduce strand rejection and D-loop reversion, favouring BIR in rad9Δ cells. The interplay between Sgs1 and Rad9 in DSB resection[7–9] could affect these results. Therefore, we tested DSB resection in

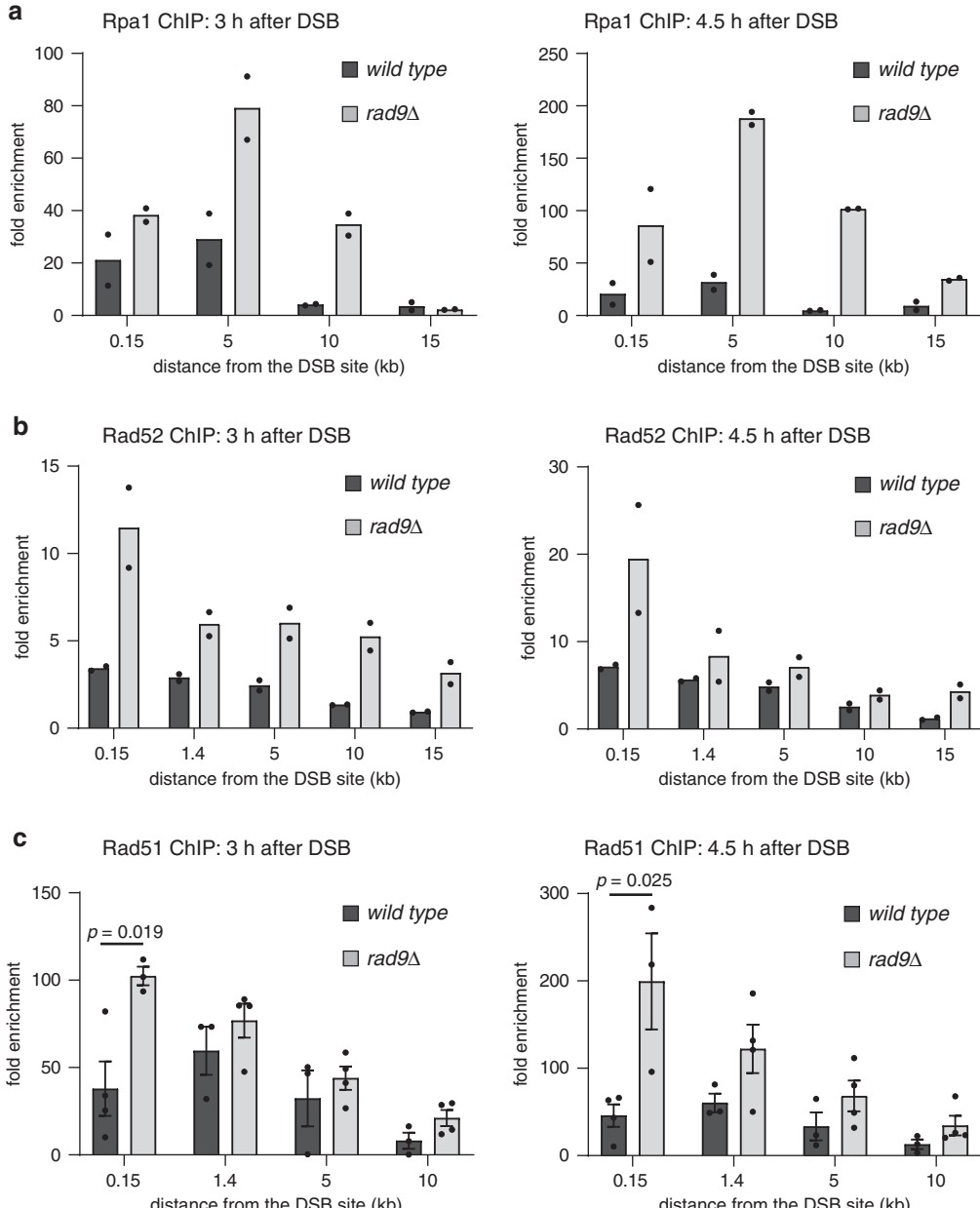

**Fig. 3 Rad9 limits Rpa1, Rad52 and Rad51 hyper-loading at a DSB.** ChIP analysis at the indicated positions from the cut site in JKM139 background, at 3 and 4.5 h following DSB induction, for the binding of **a** Rpa1-Myc ($n = 2$, biologically independent experiments), **b** Rad52-HA ($n = 2$, biologically independent experiments) and **c** Rad51 ($n = 3$ for $rad9\Delta$ and $n = 4$ for wild-type biologically independent samples). The indicated strains were blocked in G2/M with nocodazole. Data are mean ± SEM wherever $n = 3$ or more. Statistical analysis was done using unpaired two-tailed Student's $t$-test. See accompanying Source data file.

$sgs1\Delta$ and $sgs1$-FD mutants. The amount of ssDNA near the DSB ends was similar between the wild-type and $rad9\Delta$ cells (Supplementary Figs. 2a and 3a). Although the $SGS1$ deletion slightly reduced DSB resection speed alone and when combined with $RAD9$ deletion in the JKM139 background (Supplementary Fig. 2a-c), the $sgs1$-FD allele did not alter resection in both the JKM139 and JRL092 backgrounds (Supplementary Figs. 2a-c and 3a). Moreover, the $sgs1$-FD $rad9\Delta$ double mutant showed the same resection rate as the single $rad9\Delta$ in the JKM139 background (Supplementary Fig. 2a–c).

These results, together with the viability of the $sgs1$-FD $rad9\Delta$ cells in the BIR assay (Fig. 5h), confirm that the kinetics of DSB resection is not directly responsible for the severe BIR defect of the JRL092 $rad9\Delta$ cells.

**Limiting Sgs1-Mph1 by Rad9 promotes CO in ectopic DSB repair.** Since our data in Fig. 1 also showed reduced levels of CO and long-tract GC in $rad9\Delta$ cells, we speculated that the Rad9-mediated inhibition of the strand rejection might promote the CO events in gene conversion in addition to BIR repair. To investigate this aspect in more detail, we used a DSB-induced ectopic gene conversion (eGC) system (tGI354 background)[23,30], which is coupled to DDC activation[31]. This assay allowed us to detect both the CO and NCO repair products by Southern blot and PCR with specific oligonucleotides (Fig. 6a). Notably, consistent with their role in D-loop rejection, the deletion of $SGS1$ and $MPH1$ greatly increase COs in this background[23,25]. First, we found that the $RAD9$ deletion caused a mild reduction of the wild-type and $mph1\Delta$ $sgs1\Delta$ strains survival upon plating the cells in galactose to

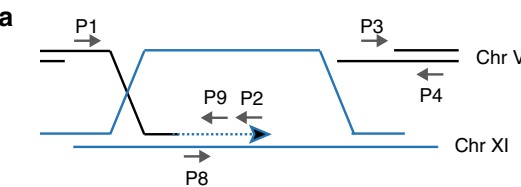

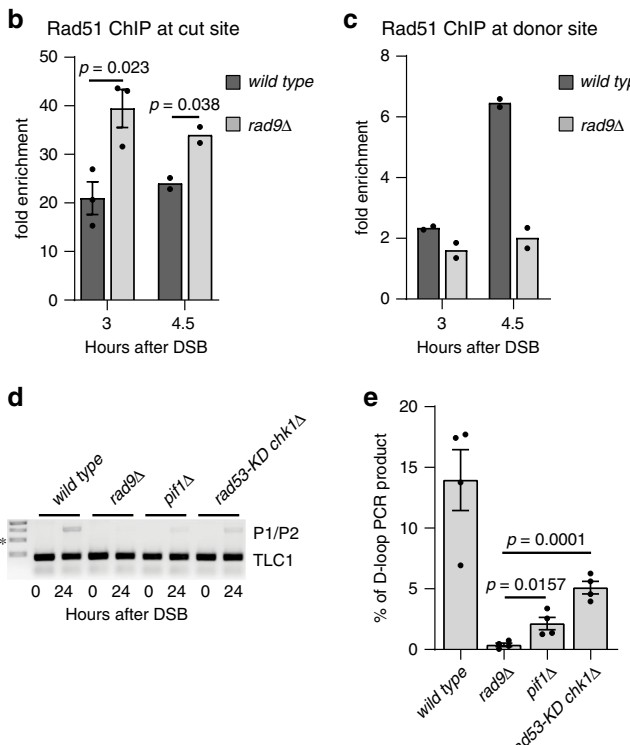

**Fig. 4 Rad9 favours Rad51 at donor site and D-loop extension in BIR.**
**a** Scheme of the PCR-based assay to test D-loop extension and ChIP in JRLO92 background. Rad51 enrichment at the cut site on chromosome V (**b**) at 3 ($n = 3$, biologically independent experiments) and 4.5 h ($n = 2$, biologically independent experiments), after DSB induction in the indicated JRLO92 strains. Rad51 enrichment at the donor site on chromosome XI (**c**) at 3 and 4.5 h after DSB formation in the indicate JRLO92 strains ($n = 2$, biologically independent experiments). In both (**b**) and (**c**) the cells were blocked in G2/M with nocodazole. **d** PCR-based assay to test D-loop extension in the indicated JRLO92 strains, (*) indicates the 1000 bp band of the 1 kb DNA ladder (NEB). **e** Quantification of the D-loop extension in four independent experiments as in (**d**). All the data in the figure are presented as mean ± SEM wherever $n = 3$ or more. Statistical analysis was done using unpaired two-tailed Student's *t*-test. See accompanying Source data file.

induce the HO-DSB in this background (Fig. 6b). Then, by PCR analysis of a large number of survivors chosen at random, we found that the CO events were less frequent in *rad9Δ* (2 of 80 colonies analysed) versus wild-type cells (6 of 60 colonies analysed). Strikingly, the deletion of *SGS1* and *MPH1* rescued the CO events of the *rad9Δ* cells (21 of 99 colonies analysed) (Fig. 6b). We also monitored DSB repair by Southern blot in cells blocked in G2/M phase by nocodazole treatment. In agreement with the previous analysis of the survivors, we found that CO events were particularly reduced in *rad9Δ* cells in Sgs1 and Mph1 dependent manner (Fig. 6c, d). These results support our

hypothesis that Rad9 promotes CO events during eGC in tGI354, likely limiting strand rejection by Sgs1 and Mph1 in D-loop maturation.

## Discussion

Over the years, several studies including ours demonstrate that Rad9, in addition to its role in DDC signalling, acts as a physical barrier at DSBs that limits the nucleolytic processing of DNA ends[32]. Importantly, these discoveries in yeast greatly contributed to understanding the roles of 53BP1 in genome stability in higher eukaryotes[11]. In this study, using different genetic assays in yeast, we show that Rad9 is also important for HR subpathway choice, providing experimental evidence of an unprecedented role of Rad9 and the DDC in controlling the fate of HR repair. Starting with a DSB repair assay in diploids[17], we found that cells lacking Rad9 show relevant increase in short-tract GC accompanied by reduced levels of long-tract GC, CO and BIR with respect to the wild type. Considering that long-tract GC, CO and BIR outcomes require extensive DNA synthesis and the formation of complex DNA structures, an explanation for their reduction in the DDC defective *rad9Δ* mutant would be that cells divide before the synthesis of long DNA tracts and the formation of stable Holliday junctions. Indeed, in a recent study with a disomic yeast strain in which a DSB can be induced on a second truncated copy of the chromosome III, *RAD9* deletion was shown to cause BIR defects associated with increased chromosome loss events, which were partially rescued by keeping the cells blocked in G2/M with nocodazole[14,15]. However, in the diploid assay we did not observe higher chromosome loss in the *rad9Δ* cells with respect to wild type. Although further investigations are needed to address this discrepancy, a possible explanation is that the disomic yeast may deal better with the loss of the broken chromosome than the diploid strain. Nevertheless, *rad9Δ* cells had slightly reduced viability respect to the wild type in our diploid assay, suggesting the intriguing possibility that DSB repair events could be trapped in lethal intermediates.

Additional experiments done using specific haploid backgrounds set up to study ectopic GC[30] and BIR[18,30], helped us to characterize the HR repair defects in *rad9Δ* cells, dissecting different steps of the recombination process.

Our results support a DDC independent function of Rad9 to prime repair DNA synthesis after strand invasion. Indeed, in contrast to a system in which a successful BIR requires the Rad53-dependent phosphorylation of Pif1 for the synthesis of about 100 kb of chromosomal DNA[16], the synthesis of 28 kb of DNA in our BIR system is less dependent on Pif1, but dramatically requires Rad9.

Strikingly, we showed that Rad9 limits the recruitment of Rpa1, Rad52 and Rad51, together with the DNA helicases Sgs1 and Mph1, at the 3′ end of the ssDNA filament. This is reminiscent of the Rad9 function in regulating DNA end resection at DSBs, where Rad9 acts as a physical barrier against the loading of helicases and nucleases[32]. We think that our results are consistent with a model in which the physical presence of Rad9, perhaps in the form of complex oligomers[7,21,33], limits the assembly of several recombination factors at the DSB, not necessarily involved in end resection. In agreement with this hypothesis, we found that the protein variants Rad9-2Ala and Rad9-7xA, which do not assemble as stable oligomers at DSBs[20,21], did not completely rescue the viability of *rad9Δ* cells upon DSB in the BIR assay. Conversely, the formation of Rad9 oligomers near a DSB is known to be higher in the absence of critical recombination factors, such as Sae2 and Slx4[7,31,34,35]. In particular, *sae2Δ* cells accumulate Rad9 at DSB ends, limiting Sgs1-dependent resection.

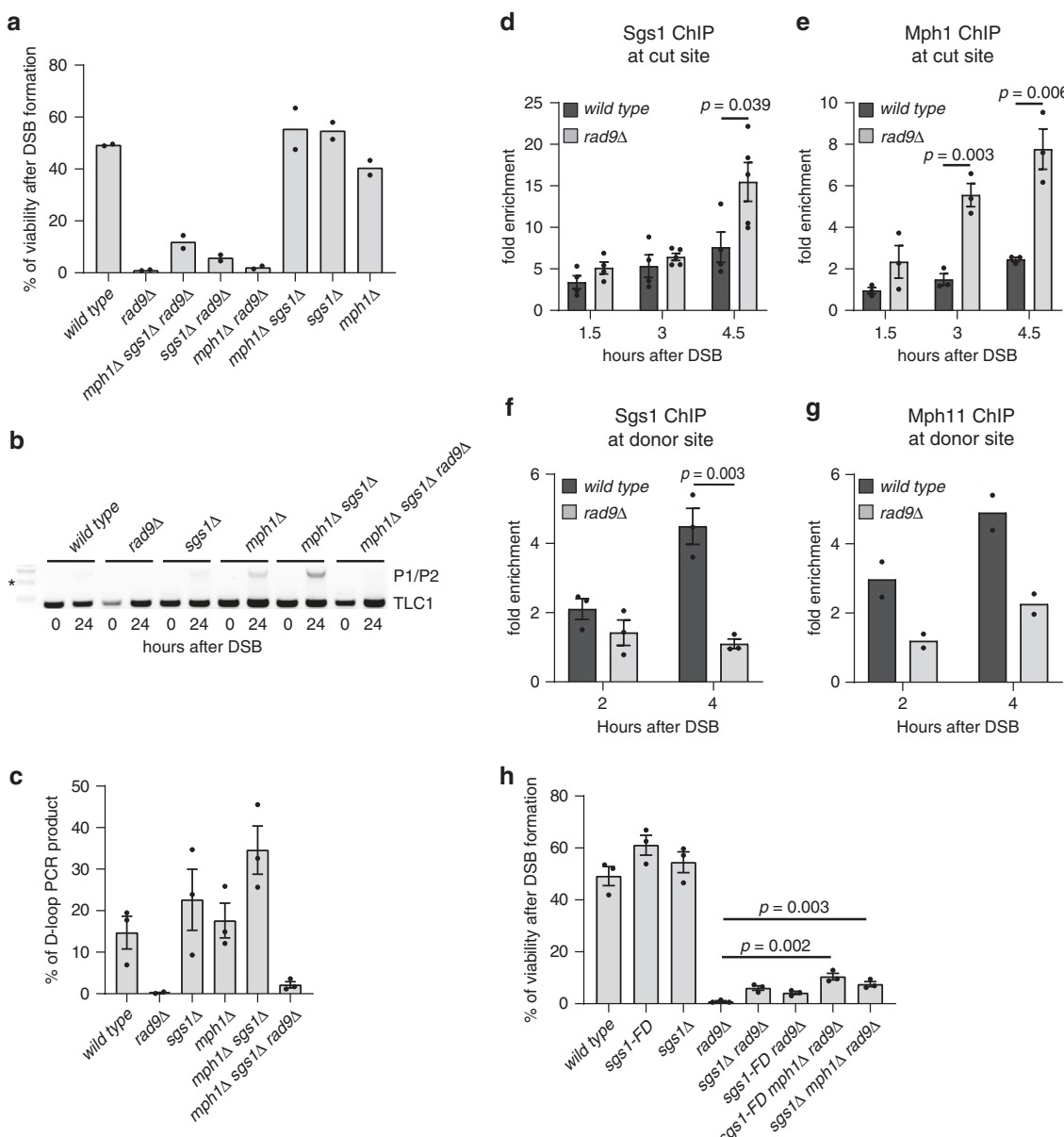

**Fig. 5 Rad9 promotes D-loop extension in BIR by limiting Sgs1 and Mph1. a** BIR efficiency measured by cell viability in the indicated JRL092 strains ($n = 2$ biologically independent experiments). **b** PCR-based assay to test D-loop extension in the indicated JRL092 strains, (*) indicates the 1000 bp band of the 1 kb DNA ladder (NEB). **c** Quantification of the D-loop extension as in (**b**) ($n = 3$ biologically independent experiments). Sgs1 (**d**) binding at the DSB site on chromosome III in the indicated JKM139 derivative strains blocked in G2/M with nocodazole (biologically independent experiments: $n = 4$ for wild type and $n = 5$ for the *rad9Δ* strain). Mph1 (**e**) binding at the DSB site on chromosome III in the indicated JKM139 derivative strains blocked in G2/M with nocodazole ($n = 3$ biologically independent experiments). Sgs1 (**f**) binding at the donor site on chromosome XI in the indicated JRL092 derivative strains blocked in G2/M with nocodazole ($n = 3$ biologically independent experiments). Mph1 (**g**) binding at the donor site on chromosome XI in the indicated JRL092 derivative strains blocked in G2/M with nocodazole ($n = 2$ biologically independent experiments). All data in panels (**d**), (**e**), (**f**) and (**g**) were normalized also on the 0 h time point. **h** As in (**a**) ($n = 3$ biologically independent experiments). All the data in the figure are presented as mean ± SEM wherever $n = 3$ or more. Statistical analysis was performed using unpaired two-tailed Student's *t*-test. See accompanying Source data file.

It is unknown whether this might also contribute to stabilizing the D-loop in *sae2Δ* cells; however, it is known that cells lacking Sae2 increase BIR efficiency[36].

Importantly, the deletion of *SGS1* and *MPH1* rescued the BIR and CO deficiencies of *rad9Δ* cells. Partial rescue was also found by the *sgs1*-FD mutation, which impairs the interaction of Sgs1 and Rad51[27], but does not interfere with DSB resection. Thus, the *sgs1*-FD allele separates two important functions of Sgs1 in HR repair, discriminating between the Sgs1 roles in D-loop rejection and DSB resection.

Overall, these data suggest that the hyper-loading of recombination factors, together with Sgs1 and Mph1, on the DSB ends might confer more dynamic mobility to the invading strand, which could be faster displaced from the homologous template during the strand invasion process in *rad9Δ* cells (Fig. 7). Several studies show that DNA synthesis requires hours for detection during BIR, while it is faster during GC repair[15,18,37–39], leading us to speculate that the D-loop intermediate is not stable enough in *rad9Δ* cells to prime DNA synthesis for BIR or long-tract GC repair, while it could sustain short-tract DNA synthesis.

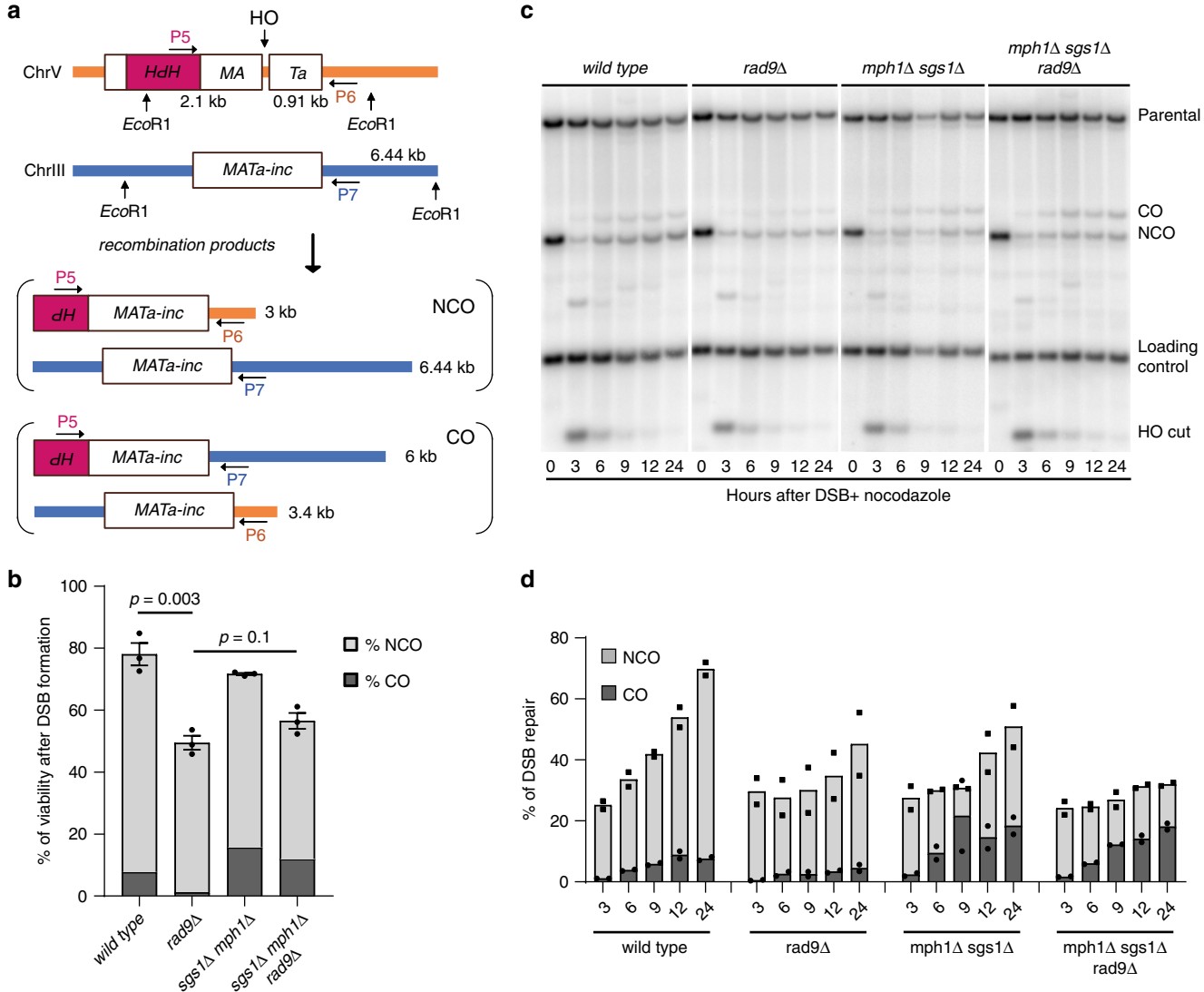

**Fig. 6 Limiting Sgs1-Mph1 by Rad9 promotes CO in ectopic DSB repair. a** Scheme of the genetic system to test ectopic gene conversion (eGC) in tGI354 background. **b** Cell viability after DSB in the indicated tGI354 derivative strains ($n = 3$ independent experiments) combined with the frequency of crossovers (CO) and non-crossovers (NCO) in the survivors measured by PCR analysis. **c** Southern blot of EcoRI-digested DNA to monitor DSB repair through eGC in the indicated tGI354 derivative strains. CO crossover, NCO non-crossover. **d** Densitometric analysis of the CO and NCO bands of two biologically independent experiments as in (**c**). All the data in the figure are presented as mean ± SEM wherever $n = 3$ or more. Statistical analysis was performed using unpaired two-tailed Student's $t$-test. See accompanying Source data file.

Consequently, COs are also much less favoured in cells lacking Rad9 (Fig. 7).

In summary, once Rad9 is recruited at a DSB, it mediates the DDC signalling, but also couples two critical regulatory events of DSB repair, the end resection and stabilization of the D-loop structure (Fig. 7). This reduces SSA and favours DSB repair through HR sub-pathways that require stable D-loops, such as long-tract GC, eGC and BIR, also increasing the frequency of CO outcomes. Accordingly, *RAD9* deletion also limits sister chromatid exchanges and promotes Rad1/XPF-dependent translocations, likely through SSA[12,13]. In line with our model, eliminating *SGS1* in *rad9Δ* cells causes dramatic levels of translocations between homeologous sequences and complex chromosomal rearrangements[14,40].

Interestingly, it has been shown that 53BP1 depletion limits GC and favours SSA[41]. Moreover, it was recently shown that 53BP1, interacting with PTIP, antagonizes HR repair by limiting Rad51 loading on a DSB in mice[42]. Based on recent discoveries that

53BP1 oligomerization in human cells drives droplet formation by liquid–liquid phase separation (LLPS)[43,44] and compartmentalization of repair sites with selective inclusion/exclusion of repair factors[45,46], it is tempting to speculate that LLPS events and droplet-like compartmentalization might regulate the Rad51 loading and maturation of functional recombinogenic filament, both in yeast and human cells. Indeed, we note that sequence analysis of yeast Rad9 shows the presence of intrinsically disordered N-terminal region (AA 1–730 out of 1309), which is well studied to drive LLPS in several proteins[47], suggesting possibly conserved mechanisms of regulation from Rad9 to 53BP1.

In conclusion, we propose that Rad9 ensures HR repair via crossover recombination only after activating DDC to restrain cell division. This reduces the risk of premature segregation of the chromosomes with DNA linkages caused by unresolved HR intermediates, which leads to the formation of anaphase bridges and deleterious chromosome rearrangements[48,49]. Indeed, 53BP1-depleted cancer cells accumulate ultrafine anaphase

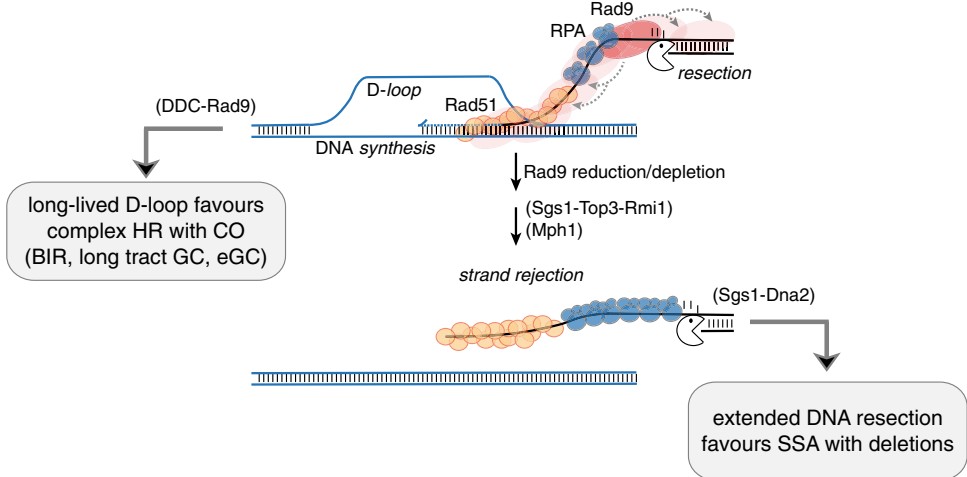

**Fig. 7 Rad9 controls the nucleoprotein filament in recombination.** After a DSB, the Rad9 oligomers assembled at the lesion limit: (i) the hyper-loading of recombination factors Rpa1, Rad52 and Rad52; (ii) the Sgs1-Top3-Rmi1 and Mph1 axes for the strand rejection, favouring long-lived D-loop and complex HR sub-pathways associated with CO (BIR, long-tract GC, eGC); (iii) the Sgs1-Dna2 axis for the long-range resection and single-strand-annealing (SSA) repair associated with deletions.

bridges, chromosome breaks and rearrangements[50]. However, according to our results, *RAD9* deletion reduces the repair events that cause extended loss of heterozygosity (LOH), a process frequently linked to tumour development. How Rad9/53BP1 finely tunes this complex regulation to preserve genome stability is a challenge for the future.

## Methods

**Yeast strains, media and growth conditions.** All the strains listed in Supplementary Table 5 are derivative of JKM179, JRL092, tGI354 and W303. To construct strains standard genetic procedures were followed[51]. Deletions and tag fusions were generated by the one-step PCR system. The *sgs1*-F1192D was obtained using a Cas9 mediated gene targeting system[52]. For the indicated experiments, cells were grown in YEP medium enriched with 2% glucose (YEP + glu), 3% raffinose (YEP + raf) or 3% raffinose and 2% galactose (YEP + raf+gal). Unless specified all the experiments were performed at 28 °C. To block cells in G2/M, 20 μg/ml nocodazole was added to the cell culture.

**Cell viability assay.** JRL092 derivative strains were inoculated in YEP + raf, grown O/N at 28 °C. The following day, cells were normalized and plated on YEP + glu and YEP + gal. Plates were incubated at 28 °C for 3 days. Viability results were obtained from the ratio between number of colonies on YEP + gal and YEP + glu. Standard error of the mean (SEM) was calculated on three or more independent experiments.

**Southern blot analysis.** Physical analysis of DSB repair kinetics during BIR and ectopic gene conversion was performed with DNA samples isolated at different time points from HO induction. Purified genomic DNA was digested with the appropriate restriction enzyme/s, probed with a specific $^{32}$P-labelled probe and scanned with a Typhoon Imager (GE healthcare). For repair analysis through BIR using JRL092 background, genomic DNA was digested with *Ava*I enzyme and separated on a 0.8% agarose gel. Southern blotting was done using a *CAN1* probe; the % of BIR repair has been calculated using the donor band as a loading control[18]. For ectopic recombination using tGI354 background, genomic DNA was digested with *Eco*RI enzyme and separated on a 0.8% agarose gel[31]. Southern blotting was done using a 1000 bp *MATa* probe. The DSB repair has been calculated by normalizing the DNA amount using a DNA probe specific for *IPL1* gene (unprocessed locus). Densitometric quantification of the band intensity was performed using the ImageJ software. The SEM was calculated on three or more independent experiments.

**ChIP analysis.** ChIP analysis was performed as described in ref. [7] with slight modifications. Briefly, cells were grown to log phase in YEP + raf and arrested in G2/M with 20 μg/mL nocodazole wherever indicated before addition of galactose to a final concentration of 2%. Cells were sampled before addition of galactose (0 h) and at time points after DSB induction as shown in respective figures. Crosslinking was done with 1% formaldehyde for 5 min (Myc or HA tagged proteins) or for 30 min (Rad51). The reaction was stopped by adding 0.125 M Glycine for 5 min. Immunoprecipitation was performed by incubating the samples with Dynabeads

Protein G (Thermo Fisher Scientific), pre-conjugated with 5 μg of 9E10 anti-Myc antibody or 12CA5 anti-HA antibody or 3 μg of anti-Rad51 antibody (PA5-34905, Thermo Fisher Scientific) for 2 h at 4 °C. Whole chromatin extract (Input) and immunoprecipitated DNA were analysed by quantitative PCR, using a Bio-Rad CFX connect, or droplet digital PCR (ddPCR), using a Bio-Rad QX200 droplet reader. For JKM139 derivative strains, several oligonucleotides have been designed at specific distance from the DSB to measure enrichment of Rpa1-HA, Rad51, Rad52-HA, Sgs1-MYC, Mph1-MYC. In JRL092 derivative strains, enrichments of Rad51, Sgs1 and Mph1 at the donor site have been evaluated with oligonucleotides on *CAN1* locus on chromosome XI. For enrichment of Rad51 at cut site, oligonucleotides at 0.2 kb from DSB have been designed. For ChIP analysis in JKM139 background, *KCC4* locus on chromosome III or *CAN1* locus on chromosome V have been used as control uncut locus; for ChIP analysis in JRL092 background, *KCC4* has been used as control uncut locus.

The oligonucleotides used are listed in Supplementary Table 6. Data are presented as fold enrichment at the HO-cut site (at the indicated distance from the DSB) or at the donor site, over that at control uncut locus, after normalizing to the corresponding input samples. ChIP data were normalized on the 0 h time point wherever specified. SEM was calculated on at least three independent experiments.

**D-loop extension analysis.** To measure the DNA synthesis after D-loop formation during BIR we adopted a strategy that was described in ref. [18]. In our experiments, suitable dilution of genomic DNA (~ 5–10 ng) at 0 and 24 h after DSB formation were PCR amplified within the linear range (~28 cycles or till *TLC1* PCR signal was not as saturation) and early BIR products were identified with primers P1 (specific to Chr. V) and P2 (specific to donor on Chr. XI). All PCR products were subjected to electrophoresis in 1% agarose, stained with ethidium bromide and quantified using Bio-Rad Image Lab software. The percentage of BIR product was determined by dividing the BIR product signal to that amplified from independent locus (*TLC1*) on Chr. II from the same input and under same conditions. See a scheme in Fig. 4a and Supplementary Table 6.

**Quantitative analysis of DSB end resection by real-time PCR.** Quantitative PCR (qPCR) analysis of DSB resection was performed as described[19]. Briefly, genomic DNA was extracted at indicated time points after the formation of a HO-induced DSB. Several oligonucleotides were designed to assess amount of ssDNA at specific distance from the DSB. The genomic DNA was digested with *Rsa*I enzyme (NEB) that cuts inside the amplicons at several distances from the DSB and only if DNA resection has reached the distance at which the restriction site is present, we will obtain amplification by qPCR with the selected oligonucleotides. A region on *KCC4* gene on chromosome III that is not cut by *Rsa*I enzyme served as control. The oligonucleotides used are listed in Supplementary Table 6. The qPCR reactions were performed on both digested and undigested templates using S to S Quantitative Master Mix 2X SYBR Green (Genespin) with the Bio-Rad CFX connect qPCR system. The ssDNA percentage over total DNA was calculated using the following formula: %ssDNA = $\{100/[(1 + 2^{\Delta Ct})/2]\}/f$, in which $\Delta Ct$ values are the difference in average cycles between digested template and undigested template of a given time point and $f$ is the HO-cut efficiency measured by qPCR. The stability of the 3′ ssDNA filament was measured in JKM139 and JRL092 derivative strains at indicated time points by comparing the undigested DNA content at DSB with respect the uncut locus *KCC4*, normalizing the values to their ratio at time 0 h (no DSB).

**DSB-induced recombination assay**. Homozygous diploids of the wild-type and *rad9*Δ strains derived from LSY2205-11C and LSY2543 (W303 background) were grown in YEP + raf overnight at 28 °C. Cells were plated on YEP + glu and YEP + gal plates to calculate plating efficiency and ensure *ADE2* auxotrophy before I-*Sce*I induction. Separately, 2% galactose was added to the cell culture to induce I-*Sce*I. After 1.5 h of induction, cells were plated on YEP + glu plates and incubated for 2–3 days at 28 °C. For experiments performed in G2/M arrested conditions, cells were treated with 2.5 μg/ml nocodazole and were maintained in the block for additional 8 h after DSB induction, before plating.

Percentage of red (the class of two long-track conversions), white (the class of two short-track conversions) and red/white (the class of one long- and one short-track conversion) was determined only for recombinant colonies. All the plates were replicated on YEP + glu plates containing Hygromycin, Nourseothricin and Synthetic Complete (SC) medium-trp, SC-met, SC-ade, SC-ade+raf, SC-ade+gal plates to determine the percentage of recombinant colonies and distinguish the NCO, CO and BIR events in each sub-class of the colonies[17]. Chromosome loss events were monitored by counting the number of recombinant colonies growing on SC-met and SC-trp plates.

**Statistical analysis**. Statistical analysis was performed using Microsoft Excel or Prism software. *P*-values were determined by an unpaired two-tailed *t*-test. No statistical methods or criteria were used to estimate sample size or to include or exclude samples.

**Reporting summary**. Further information on research design is available in the Nature Research Reporting Summary linked to this article.

## Data availability

All data are in the paper and supplementary information. The source data underlying Figs. 2b–d, 3a–c, 4b–e, 5a–h, 6b–d and Supplementary Figs. 2a–d, 3a, b, 4a, 5a–c are provided as a Source data file. All data are available from the authors upon reasonable request.

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

## Acknowledgements

We thank Lorraine S. Symington and James E. Haber for generously providing yeast strains. We thank Gerard Mazon for useful suggestions for using LSY2205-11C/LSY2543 background. We are grateful to James E. Haber and all the members of our laboratory for helpful discussions. We thank Christopher P. Caridi for reading the paper. This work was supported by grants to A.P. from Associazione Italiana Ricerca sul Cancro (AIR-C_IG19917, AIRC_IG15488) and from Ministero Istruzione Università e Ricerca, MIUR (PRIN-2015LZE994). M.F. was supported by a fellowship from Fondazione Gabriella Dolfin Voyasidis-Accademia Nazionale dei Lincei. C.C.R. was supported by a fellowship from Università degli Studi di Milano (Assegno di ricerca –Tipo A).

## Author contributions

M.F. and A.P. conceived the idea. M.F. performed cell viability assay in Figs. 2d, 5a and Supplementary Fig. 4; ChIP analysis in Figs. 3 and 4c; D-loop extension analysis in Figs. 4d, e and 5b, c; contributed to Southern blot and DSB resection analyses in Figs. 2b, c, 6c, d, Supplementary Figs. 5a, b and 2. C.C.R. performed DSB repair assay with the diploid system in Fig. 1 and Supplementary Tables 1, 3; contributed to D-loop extension, Southern blot and DSB resection analyses in Figs. 2b, c, 4d, e, 5b, c, 6c, d, and Supplementary Fig. 2. S.L. performed ChIP analyses in Figs. 4b, 5e–g and Supplementary Fig. 5c; DSB resection analyses in Supplementary Fig. 4; contributed to Southern blot analysis in Supplementary Fig. 5a, b. M.Y.V. performed DSB repair assay with the diploid system in Fig. 1 and Supplementary Tables 1–4; cell viability assay in Figs. 5h and 6b; PCR analysis of the survivors in Fig. 6b. M.F., C.C.R. and A.P. wrote the paper. M.F., C.C.R., S.L., M.Y.V. and A.P. revised the text. A.P. supervised and coordinated all aspects of the work.

## Competing interests

The authors declare no competing interests.
