## [Peer Review File · Nature Communications]

Reviewers' comments:

Reviewer #1 (Remarks to the Author):

The authors uncover a new role for Rad9/53BP1 in double-strand break repair that promotes crossover recombination by stabilizing D-loops at the invading 3' single strand annealed to the donor template and thereby inhibiting the DNA helicases Sgs1 and Mph1 which can act to undo the D-loop intermediate. These findings add a new layer to the complex regulation of double-strand break repair. As the gene products studied here are conserved, the findings likely have implications for DNA break repair in mammalian cells.

The crux of the study and indeed the starting point is the use of a color reporter assay that interprets the colony color outcomes following DNA break induction as the length of a gene conversion tract. To those conversant in the recombination models, this is obvious, but to other readers this might be totally opaque. Also perhaps not understood is the fact that each break event in a cell results in two recombination repair events if the break is induced or repaired in S/G2 and one if everything happens in G1, a scenario unlikely to occur. These aspects of the reporter system need to be explained, even if a supplementary figure is needed. In this section BIR also needs to be explained, what it is, what its outcome would be in terms of red and white colonies and why.

In a similar vein, Figure 4d needs more explanation and a better legend. Even the color of the boxes, red versus grey, are not explained. Here it would be useful to point out what are the potentially deleterious products in terms of genome instability or loss of heterozygosity as presumably Rad9/53BP1 would promote crossover recombination only under certain circumstances. Finally, is there any regulation of Rad9 function in checkpoint signaling and DSB processing versus a role in stabilizing D-loops?

Some of the English needs editing.

Reviewer #2 (Remarks to the Author):

In this study, the authors investigate the role of budding yeast Rad9 (53BP1 in mammalian cells) in regulating homology-dependent repair outcomes using several assay systems. First, by measuring DSB-induced repair between chromosome homologs in diploid cells, the authors show an increase in short tract non-crossover (NCO) recombinants and a corresponding reduction in crossover (CO) and BIR products in the *rad9* mutant. The reduced BIR in the *rad9* mutant was verified using another assay that specifically detects BIR, and the authors show that the *rad9* defect is not due solely to loss of DNA damage signaling/cell cycle arrest or increased end resection. Interestingly, the *rad9* mutant shows normal Rad51 recruitment near the DSB but a defect in Rad51 association with the donor template, suggesting Rad9 controls strand invasion or D-loop stability. Previously, Mph1 and Sgs1 had been shown to suppress BIR, presumably due to their D-loop resolution activities. *mph1* and *sgs1* mutations partially suppress the *rad9* BIR defect, which the authors suggest is due to Rad9 suppression of Mph1 and Sgs1 binding near the initiating DSB. Finally, the authors show that Rad9 suppresses COs using an ectopic recombination assay and this suppression appears to be mediated by Mph1 and Sgs1.

The finding that Rad9 influences recombination outcomes will be of considerable interest to the DNA repair community. However, there are some inconsistencies in the results presented that need to be addressed.

1. Vasan et al (2014) reported a smaller decrease in BIR in the *rad9* mutant than observed here and also a partial suppression of the *rad9* defect by *sgs1*. These results should be discussed. Vasan used an assay where there is much more extensive homology between the broken

chromosome and donor duplex. One potential concern with the assays used in Figures 2-3 is that there is limited homology. If resection is much more extensive in these strains, the 3' ends could be degraded removing the homology needed for strand invasion. I suggest the authors measure by PCR whether there is 3' end loss 6 hr after HO induction in the YLR strain at the time they report a loss in Rad51 binding to the donor.

2. Does the *rad9* mutant show equivalent plating efficiency to WT after DSB induction the diploid assay and is there any increase in chromosome loss events? These numbers should be presented.

3. The Haber lab (Lydeard et al, 2007; 2010) previously reported a much lower percent viability of the WT strain after DSB formation using the same BIR assay as used here (20% vs 50%). Do the authors have an explanation for this difference? Also, *sgs1* and *mph1* mutations were previously shown to significantly increase the frequency of BIR using this and similar assays, in contrast to the data shown in Fig 3d.

4. How would Rad9 binding to chromatin influence Rad51 binding to the ssDNA formed by end resection? The *rad9-7xA* and *rad9-2Ala* mutants exhibit a significant rescue of the *rad9* BIR defect suggesting that chromatin binding by Rad9 is not required (Sup Fig 3).

5. I am concerned that the Rad51, Rad52 and RPA ChIP and end resection assays were performed in the JKM strain. This is a different genetic background to the W303 and YLR strains used for the genetic assays in Figures 1-3. I think it is important to measure Rad51 binding at the recipient and donors in the same (YLR) background and to show that the *sgs1*-FD mutant is proficient for end resection in the YLR strain.

6. Presumably *Sgs1* and *Mph1* are acting on the D-loop intermediate. I don't see the relevance of measuring *Sgs1* and *Mph1* binding to the broken chromosome. Are *Mph1* and *Sgs1* bound to the donor chromosome and if so is their binding influenced by Rad9? Because the *rad9* mutant exhibits faster resection, the increase in *Mph1* and *Sgs1* binding could simply be due to the increase in ssDNA substrate available. Is *Mph1* binding increased in the *rad9 sgs1* double mutant as compared to *sgs1*?

7. In the assay shown in Figure 1, more NCOs were recovered from the *rad9* mutant than from WT; however, in the ectopic assay (Figure 4) NCOs are reduced in the *rad9* strain. If Rad9 limits access of the strand invasion intermediate to *Sgs1* and *Mph1* one would expect an increase in NCOs in this assay. These data do not fit with their model. Because repair is measured only by physical assay in Fig 4, the bands labeled as the CO products (Fig 4b) could instead be due to BIR. The percent survival in this assay should be shown and survivors need to be analyzed for CO and NCO outcomes.

Minor comments:

Line 24 and elsewhere: replace "tracks" with "tract".

What is the evidence for Rad9 limiting MMEJ?

Figure 2a: I would have expected higher RPA binding 15 kb from the DSB in the *rad9* mutant. Is there any value to showing ChIP values 15 kb from the DSB in parts a and b?

Figure 3i: the *sgs1* null should be directly compared with *sgs1*-FD and p values presented.

Figure S1. Why are these data shown again or is this an independent trial?

Reviewer #3 (Remarks to the Author):

Ferrari et al. utilized several recombination assays in baker's yeast to assess roles for Rad9 in regulating DNA repair events. First, they tested in diploids the effect of the *rad9Δ* mutation in a mitotic recombination assay (Ho et al. Mol Cell 40:988) that assesses the repair of induced double-strand breaks through gene conversion, crossing over, and break-induced replication (BIR). They observed in *rad9Δ* mutants an increase in short gene conversion tracts (white colonies) and crossover events, and a modest, but statistically insignificant, decrease in BIR.

In a physical assay performed in haploid cells that measures BIR products, *rad9Δ* mutants displayed a strong defect that appeared independent of nocodazole treatment and was different

from that seen in *pif1Δ* and *rad53-KD chk1Δ* mutants. The later observation, in conjunction with other assays, suggested that the *rad9Δ* phenotype in this assay was unrelated to DDC signaling. They then performed ChIP analysis and showed that Rpa1, Rad52 and Rad51 displayed higher levels near an induced double-strand break in *rad9Δ* mutants following DSB formation, but reduced Rad51 localization at a donor site. This work was supported by the use of a D-loop stabilization assay developed by the Heyer lab. The authors found that *rad9Δ* mutants displayed a defect in D-loop stabilization that was apparently suppressed by *mph1Δ sgs1Δ*, suggesting a coordination between the two.

Lastly, they used a physical ectopic gene conversion assay developed by the Haber lab to monitor in haploids repair events at MAT. They found that repair was reduced in *rad9Δ*, with crossover but not noncrossover events suppressed in the *mph1Δ sgs1Δ* mutant.

Based on the above observations Ferrari et al. propose a novel role for Rad9 in promoting crossover formation by limiting Sgs1 and Mph1 helicases. In their model they suggest that Rad9 plays a role in stabilizing annealing between 3' ssDNA generated at resected DSB and a donor template. Such an activity would promote, in coordination with an activated DNA damage checkpoint, the formation of stabilized D-loops that result in repair through homologous recombination pathways.

Comments

This an intriguing set of observations obtained from a variety of different assays (some in diploids, others in haploids) that provide a model to explain how Rad9 coordinates its role in the DNA damage checkpoint with a separate role to direct repair through homologous recombination pathways. My major concerns, outlined below, have to do with documentation-it was very difficult for this reader to determine exactly what strains were used in which assay. Also, I felt that the abstract overstated the overall conclusions and in many cases differences that were observed appeared just on the edge of statistical significance. More specific comments are below.

1. The Abstract feels disjointed, at least to this reader. The first six lines appear superfluous, and could be better used to explain to the reader the premise of why it would be important to couple DDC activation with specific homologous recombination pathways. Also, the conclusions of the abstract (We show that Rad9 stabilizes....) feel overstated relative to the more conservative proposal made in the discussion. I am impressed with the variety of assays performed, but at times the authors could do a better job linking the logic for using them and also provide some information indicating that they are measuring the same molecular process. I know that the authors are aiming for brevity, but in doing so it makes it harder for the reader to be confident of the results.

2. For the Ho et al. assay, did the authors distinguish between the types of red colonies obtained? In Ho et al. the authors use a reinduction assay to distinguish between red colonies that were due to long tract gene conversions from those that had not been induced by I-SceI. This seems important to me because red colonies in wild-type represent the majority class. Also, did the authors consider performing the I-SceI induction in the presence of nocodazole? This would help eliminate issues concerning effects due to the DNA damage checkpoint.

3. Figure 1a-should also show cartoons for crossover and BIR outcomes-these could be put into the supplement if space is a concern.

4. In Figure 1e. Did the authors determine if BIR product could be rescued in the *rad9Δ sgs1Δ mph1Δ* mutant? If this was the case it would provide some continuity between assays (e.g. results seen in Figure 3d and 3f).

5. Statistical significance in some cases appears to be on the edge for results that are highlighted in the manuscript. For example, in Figure 3D, $p=0.045$ for the difference in viability after DSB

formation between *rad9Δ* and *mph1Δ sgs1Δ rad9Δ*, and in Figure 1C, the difference in the percent of BIR events between wild-type and *rad9Δ* was 0.064. Importantly, the authors make a big deal of the finding that *rad9Δ* mutants displayed a defect in D-loop stabilization that was suppressed somewhat by *mph1Δ sgs1Δ* but at least to this viewer the differences seem subtle (Figure 3 F and I, with no statistical differences shown). In Figure 3I, are the error bars as small as they appear to this reader?

6. It was very hard for me to assess exactly how the resection assay shown in Supplemental Figure 2 was performed. There is little information provided with respect to which strains were used, as well as the methodologies involved to determine specific distances from the double-strand break. At the very least they can be documented in greater detail in the Supplement.

7. The ChIP assays also suffer from minimal detail. The authors should document the controls that indicate specificity of the antibodies as well as the specific strains in which these experiments were formed in. The authors indicate JKM strains, but this is only briefly shown in the Supplementary Table 2, without a direct visualization of the assay.

Response to Referees Letter

We are sending you a revised version of the Manuscript No. NCOMMS-19-23713-T, entitled '*Rad9/53BP1 promotes DNA repair via crossover recombination by limiting the Sgs1 and Mph1 helicases*'.

First of all, we wish to thank the three Reviewers for the time spent on our submission. Indeed, we received very detailed and valuable suggestions that we believe contributed to improve substantially the revised manuscript.

Below is a detailed discussion of the various criticisms raised and of the modifications we made at the original manuscript.

Kind Regards,

Achille Pelliccioli

Reviewers' comments in Blue, our response in Black:

Reviewer #1:

The crux of the study and indeed the starting point is the use of a color reporter assay that interprets the colony color outcomes following DNA break induction as the length of a gene conversion tract. To those conversant in the recombination models, this is obvious, but to other readers this might be totally opaque. Also perhaps not understood is the fact that each break event in a cell results in two recombination repair events if the break is induced or repaired in S/G2 and one if everything happens in G1, a scenario unlikely to occur. These aspects of the reporter system need to be explained, even if a supplementary figure is needed. In this section BIR also needs to be explained, what it is, what its outcome would be in terms of red and white colonies and why.

ANSWER: In the revised manuscript we expanded the description of the recombination assay in the diploid system, also adding additional schemes in Supplementary Fig. 1. As also requested by Reviewer's #3, we added the results obtained in cells blocked in G2/M by nocodazole, before the induction of I-*SceI*. The new results in G2/M blocked cells are similar to the results obtained in asynchronous cells (Fig. 1b,c; Supplementary Table 1-4). We hope these data and other controls that we added in the Supplementary Table 1 and 3 (the re-induction assay and the number of chromosome loss events) will help to address most of the points raised by the Reviewers for the results in the diploid system.

Figure 4d needs more explanation and a better legend. Even the color of the boxes, red versus grey, are not explained. Here it would be useful to point out what are the potentially deleterious products in terms of genome instability or loss of heterozygosity as presumably Rad9/53BP1 would promote crossover recombination only under certain circumstances.

ANSWER: We modified the boxes in the model (now in Fig. 7). We have also extensively re-written the Discussion section, trying to explain better the novelty our results suggest regarding the role of Rad9 to promote the crossover recombination DNA repair.

Finally, is there any regulation of Rad9 function in checkpoint signaling and DSB processing versus a role in stabilizing D-loops?

ANSWER: This is a very important and intricate conceptual question asked by the reviewer. Here we propose that the nucleation of Rad9 oligomers near to the DSB site controls the mobility of the nucleoprotein recombinogenic filament and its stable association to the donor site. Moreover, we provide evidence that Rad9 controls BIR through a mechanism that is not completely dependent upon Rad53 and Chk1. However, we predict that the upstream checkpoint kinases Tel1/ATM and/or Mec1/ATR will be potentially involved in the process, not only regulating the Rad9 binding at the DSBs, but also by the regulation of critical recombination factors. Indeed, it is known that DDC controls through phosphorylation several recombination factors which are implicated in DSB resection and HR repair, and could be also potentially implicated in the novel regulatory phenomenon that we described here. We think that the elucidation of the network of these phosphorylation events by Mec1 and Tel1 will be important to address how Rad9 exerts its multiple functions at DSBs, controlling end

resections, D-loop stability/maturation and DDC signalling. Future investigation will provide appropriate answers to the interesting question raised by the Reviewer.

Some of the English needs editing.

ANSWER: We extensively revised the text of the manuscript and we carefully checked spelling and grammar issue. A native English speaker, the colleague Dr. Christopher P. Caridi kindly edited grammar of the manuscript.

Reviewer #2

Vasan et al (2014) reported a smaller decrease in BIR in the *rad9* mutant than observed here and also a partial suppression of the *rad9* defect by *sgs1*. These results should be discussed. Vasan used an assay where there is much more extensive homology between the broken chromosome and donor duplex. One potential concern with the assays used in Figures 2-3 is that there is limited homology. If resection is much more extensive in these strains, the 3' ends could be degraded removing the homology needed for strand invasion. I suggest the authors measure by PCR whether there is 3' end loss 6 hr after HO induction in the YLR strain at the time they report a loss in Rad51 binding to the donor.

ANSWER: We thank the Reviewer for value suggestions. In the revised manuscript we provided experimental evidence by qPCR and ChIP analysis that the ssDNA recombinogenic filament, both in the JKM139 and JRL092 backgrounds, is stable several hours in the wild type and cells lacking Rad9 (Fig. 4c; Supplementary Fig. 2d; Supplementary Fig. 3b; and related source data files). In the revised MN, throughout the text we also discussed the results by Vasan et al. with more details.

Does the *rad9* mutant show equivalent plating efficiency to WT after DSB induction the diploid assay and is there any increase in chromosome loss events? These numbers should be presented.

ANSWER: In the revised MN we added these important numbers and related discussion (Supplementary Tables 1-4).

The Haber lab (Lydeard et al, 2007; 2010) previously reported a much lower percent viability of the WT strain after DSB formation using the same BIR assay as used here (20% vs 50%). Do the authors have an explanation for this difference? Also, *sgs1* and *mph1* mutations were previously shown to significantly increase the frequency of BIR using this and similar assays, in contrast to the data shown in Fig 3d.

ANSWER: We do not have a clear explanation for the discrepancy between our results and the original results published by the Haber lab regarding the viability of the JRL092 derivatives cells. Indeed, this could be also the case of the results with the *mph1 sgs1* mutant. Perhaps in agreement with the literature, we observed higher amounts of D-loop PCR product (Fig. 5b, c and accompanying source data files) and BIR repair gel band (Supplementary Fig. 5a, b and accompanying source data files) in the *mph1 sgs1* mutants. However, as correctly pointed out by the Reviewer, the viability was not significant higher respect the wild type cells (Fig. 5a, h and accompanying source data files).

In long experience of utilization of several genetic systems set up by different laboratories (Haber Lab included) to study DNA repair, we often noted some variations in term of kinetic of repair and viability, when performing the experiment in our lab. One possibility is that cells could be sensitive to small difference in composition of the culture media, which potentially affects the repair efficiency and cell viability. Consistent with this, sometimes we observed significant variation in these parameters when we have changed the company from which we purchased the reagents for the preparation of the cell culture media. Moreover, slight variation of other unpredictable experimental parameters might contribute too. Therefore, we carefully evaluate the addition of appropriate reference controls in all the experiments. In general, we are confident that the overall trend of our results appears consistent with the literature.

How would Rad9 binding to chromatin influence Rad51 binding to the ssDNA formed by end resection? The *rad9-7xA* and *rad9-2Ala* mutants exhibit a significant rescue of the *rad9* BIR defect suggesting that chromatin binding by Rad9 is not required (Sup Fig 3).

ANSWER: We kindly ask the Reviewer to consider that we have discussed the results of the supplementary Fig. 4 (previous Fig. 3) in a different way. Indeed, we showed that both the *Rad9-7xA* and *Rad-2Ala* only partially rescue the viability of cells lacking Rad9 in the BIR assay. As such, we think that physical binding of Rad9 oligomers at the DSB site is crucial to control Rad51 and the other

recombination factors. Additionally, we have discussed the phase separation as new biophysical property of 53BP1 described in several recent publications, which might shed light on this mechanism. Recent work from Nussenzweig lab, also presented similar mechanisms of 53BP1 limiting HR by counteracting Rad51 in mice (Callen et al. *Mol Cell*, 2020). As we discussed in the revised manuscript (please see the Discussion session), the sequence analysis of yeast Rad9 shows the presence of intrinsically disordered N-terminal region (AA 1-730 out of 1309), which might be involved in regulation mechanisms through phase separation, similarly to 53BP1.

I think it is important to measure Rad51 binding at the recipient and donors in the same (YLR) background and to show that the *sgs1*-FD mutant is proficient for end resection in the YLR strain.

ANSWER: We performed the requested experiments in JRL092 derivative strains (Fig. 4b, c; Supplementary Fig. 3a and accompanying source data files).

Presumably Sgs1 and Mph1 are acting on the D-loop intermediate. I don't see the relevance of measuring Sgs1 and Mph1 binding to the broken chromosome. Are Mph1 and Sgs1 bound to the donor chromosome and if so is their binding influenced by Rad9?

ANSWER: As requested, we performed the CHIP analysis at the donor site in JRL092 background for Sgs1 and Mph1. The new results indicate that the binding of both Sgs1 and Mph1 at the donor site is greatly reduced in cells lacking Rad9 (Fig. 5f, g and accompanying source data files), supporting the other results that the D-loop is unstable in *rad9* mutant.

Although we agree with the Reviewer that the analysis of the various repair parameters should be done in the same JRL092 background, in the revised manuscript we maintained also the resection and CHIP analyses that we did in the context of irreparable DSB in JKM139 background. Indeed, we explained in the text that the different repair kinetics in *rad9* mutant with respect to the wild type cells prevents unbiased comparison of the results between the different strains. Thus, we think that also the results obtained in the JKM139 derivatives are useful in our work.

Because the *rad9* mutant exhibits faster resection, the increase in Mph1 and Sgs1 binding could simply be due to the increase in ssDNA substrate available. Is Mph1 binding increased in the *rad9 sgs1* double mutant as compared to *sgs1*?

ANSWER: We provided data that the resection speed very near to the DSB site is similar in the wild type and cells lacking Rad9 (Supplementary Fig. 2a; Supplementary Fig. 3a; and accompanying source data files). Therefore, we did not perform the CHIP analysis of Mph1 in *sgs1 rad9* double mutant.

In the assay shown in Figure 1, more NCOs were recovered from the *rad9* mutant than from WT; however, in the ectopic assay (Figure 4) NCOs are reduced in the *rad9* strain. If Rad9 limits access of the strand invasion intermediate to Sgs1 and Mph1 one would expect an increase in NCOs in this assay. These data do not fit with their model. Because repair is measured only by physical assay in Fig 4, the bands labeled as the CO products (Fig 4b) could instead be due to BIR. The percent survival in this assay should be shown and survivors need to be analyzed for CO and NCO outcomes.

ANSWER: We thank the Reviewer for this important suggestion. In the revised MN, we analysed CO and NCO in the survivors by a PCR-based strategy (Fig. 6a, c and accompanying source data files). We also added the cell viability analysis for the different mutants (Fig. 6b). Indeed, the ratio between NCO and CO increases in the *rad9* mutant survivors of the ectopic GC assay in haploids (Fig. 6), in agreement with the results of the assay performed in diploid cells (Fig 1). This behaviour is not evident by the direct analysis of BIR repair products performed in G2/M blocked cells (Fig. 6d, e), likely because of different experimental conditions and/or technical issues during the Southern blotting procedure. Indeed, the normalization/quantification of the repair bands in the gels is not a trivial procedure, which might result in unwanted small variation in the analysis of the data, particularly when the intensity of the signal is low. However, we think that the Southern blot analysis is useful to show the significant reduction of CO recombination after 24 hours of DSB induction in cells lacking Rad9.

Line 24 and elsewhere: replace “tracks” with “tract”.

ANSWER: Done. We appreciate the correction.

What is the evidence for Rad9 limiting MMEJ?

ANSWER: In the revised text we eliminated this point. Thank you.

Figure 2a: I would have expected higher RPA binding 15 kb from the DSB in the *rad9* mutant. Is there any value to showing ChIP values 15 kb from the DSB in parts a and b?

ANSWER: In the revised Fig. 3 we added the complete ChIP analysis performed in the JKM139 derivatives. Indeed, after 4.5 hours of DSB induction the Rpa1 binding is slightly higher at 15 kb from the break in cells lacking Rad9 (Fig. 3a, see accompanying source data files).

Figure 3i: the *sgs1* null should be directly compared with *sgs1*-FD and p values presented.

ANSWER: We repeated the analysis, also considering a direct comparison to the *sgs1* null mutant, as requested. The new results with statistic analysis are in Fig. 5h (and accompanying source data files).

Figure S1. Why are these data shown again or is this an independent trial?

ANSWER: We apologize for the confusion. As discussed above, we combined all the related ChIP analysis in the new Fig. 3.

Reviewer #3

My major concerns, outlined below, have to do with documentation-it was very difficult for this reader to determine exactly what strains were used in which assay.

ANSWER: In the revised text and Figures, we indicated better the names of the strains used in the different experiments.

Also, I felt that the abstract overstated the overall conclusions and in many cases differences that were observed appeared just on the edge of statistical significance.

ANSWER: We agree that in certain cases the results obtained in the *rad9* mutants (particularly when combined with the *sgs1* and *mph1* mutations) differ by very low number. However, the small differences have been observed in many experimental approaches that we did. Moreover, whenever possible we also added the statistical analysis. Perhaps, part of the problem is due to the fact that our

results are related to rare intermediates of the DNA recombination repair. However, we think that the Reviewer will agree that these rare events should be taken into consideration to understand genome instability, typical of tumour cells.

The Abstract feels disjointed, at least to this reader. The first six lines appear superfluous, and could be better used to explain to the reader the premise of why it would be important to couple DDC activation with specific homologous recombination pathways. Also, the conclusions of the abstract (We show that Rad9 stabilizes....) feel overstated relative to the more conservative proposal made in the discussion.

ANSWER: According to the Reviewer's suggestion, we revised the text of the abstract.

For the Ho et al. assay, did the authors distinguish between the types of red colonies obtained? In Ho et al. the authors use a reinduction assay to distinguish between red colonies that were due to long tract gene conversions from those that had not been induced by I-SceI. This seems important to me because red colonies in wild-type represent the majority class.

ANSWER: In the revised manuscript we expanded the description of the recombination assay in the diploids. We also added the results of the re-induction assay to rule out experimental flaws (Supplementary Tables 1 and 3).

Also, did the authors consider performing the I-SceI induction in the presence of nocodazole? This would help eliminate issues concerning effects due to the DNA damage checkpoint.

ANSWER: Thank you for the important suggestion. The results obtained in the assay performed in G2/M blocked cells in the presence of nocodazole are in Fig. 1b, c and in Supplementary Tables 3 and 4. Importantly, the new results confirm the results obtained in the asynchronous cells (Fig. 1b, c; Supplementary Table 1 and 2).

Figure 1a-should also show cartoons for crossover and BIR outcomes-these could be put into the supplement if space is a concern.

ANSWER: We added the requested schemes in the Supplementary Fig. 1.

In Figure 1e. Did the authors determine if BIR product could be rescued in the *rad9Δ sgs1Δ mph1Δ* mutant? If this was the case it would provide some continuity between assays (e.g. results seen in Figure 3d and 3f).

ANSWER: According to the Reviewer's suggestion, we performed the analysis by Southern blotting for the BIR repair in the *rad9 sgs1 mph1* mutant (Supplementary Fig. 5a, b and accompanying source data files).

5. Statistical significance in some cases appears to be on the edge for results that are highlighted in the manuscript. For example, in Figure 3D, $p=0.045$ for the difference in viability after DSB formation between *rad9Δ* and *mph1Δ sgs1Δ rad9Δ*, and in Figure 1C, the difference in the percent of BIR events between wild-type and *rad9Δ* was 0.064. Importantly, the authors make a big deal of the finding that *rad9Δ* mutants displayed a defect in D-loop stabilization that was suppressed somewhat by *mph1Δ sgs1Δ* but at least to this viewer the differences seem subtle (Figure 3 F and I, with no statistical differences shown). In Figure 3I, are the error bars as small as they appear to this reader?

ANSWER: We repeated the viability assay that was originally shown in Fig. 3. The new data are shown with statistic analysis in the new panel Fig. 5h.

Please, see above our discussion regarding the significance of subtle differences found in the different mutants.

It was very hard for me to assess exactly how the resection assay shown in Supplemental Figure 2 was performed. There is little information provided with respect to which strains were used, as well as the methodologies involved to determine specific distances from the double-strand break. At the very least they can be documented in greater detail in the Supplement.

ANSWER: We expanded the analysis for DSB resection, which is now shown in both the JKM139 derivatives (Supplementary Fig 2 and accompanying source data files) and JRL092 derivatives (Supplementary Fig. 3 and accompanying source data files). The names of the yeast backgrounds used for the analysis are indicated in the legends of each figure. The qPCR-based protocol used for the DSB

resection analysis is briefly described in the Methods. Please, see that we referred to a paper (Ferrari et al 2018) in which we described the protocol with major details.

The ChIP assays also suffer from minimal detail. The authors should document the controls that indicate specificity of the antibodies as well as the specific strains in which these experiments were formed in. The authors indicate JKM strains, but this is only briefly shown in the Supplementary Table 2, without a direct visualization of the assay.

ANSWER: We briefly expanded the description of the ChIP protocol. The names of the yeast backgrounds used for the ChIP analysis and other assays described in the manuscript are now indicated in each figure.

Considering that the ChIP analysis at the donor site in JRL092 background was a new experimental setting, as suggested by the Reviewer we did a control test for the specificity of the 9E10 antibody used for the MYC-tagged Sgs1 and Mph1 variants (Supplementary Fig. 5d).

As for the Rad51 ChIP, we used the previously reported polyclonal anti-Rad51 that were already used for similar application (Graf et al. Cell, 2017, 170 (19:72-85. doi: 10.1016/j.cell.2017.06.006) and validated by Thermo Fischer Scientific. In past, we tested the specificity of 12CA5 antibodies that we used for Rpa1-HA and Rad52-HA ChIP analyses at cut site in JKM139 derivatives. Based on huge literature and long experience in our lab, they are highly specific for this application. We hope that these justifications will satisfy the Reviewer's requests.

Reviewers' comments:

Reviewer #1 (Remarks to the Author):

The authors have considered the three reviewers' comments very carefully and have performed additional experiments and have rewritten sections of the manuscript and added and modified figures. The response to the comments is good and the manuscript is much improved. It is clearer now which strains/constructs are used in which experiments, although there remains the unsatisfying finding of different rates/measurements with the same strain in different labs. It is known that media source is important, that the best source seems to be Difco and other formulations can have inhibitory effects on sporulation and cell viability. Nonetheless, the results stand in this manuscript and there are bound to be some difference between labs.

Importantly, the authors show that Rad9/53BP1 has an additional role in DSB repair and pathway choice through D-loop stabilization, targeting the Sgs1 and Mph1 DNA helices. This should have implications that extend to mammalian DSB repair pathways. A few minor comments are below.

1. line 89. I think ADE2/ade2 and ade2/ade2 should be italicized.

line 94. This is probably a difference between English and Italian verb tenses. All survivors are tested, not should be tested. The rationale for I-SceI re-induction is not stated although who do such experiments understand the reason.

3. line 178. "cells lacking RAD9". Here RAD is italicized, referring to the gene. However, I think in the case the focus is on the absence of the protein so it should be Rad9.

Reviewer #2 (Remarks to the Author):

The authors have addressed most of the issues raised at the first review. However, I still have some concerns, and suggestions for presentation of the data.

Figure 1c: It would be helpful to plot the percent of events adjusted for the cell viability, which is about 10% lower for rad9 than wild type. I suspect that with this adjustment NCOs are not increased and BIR and CO products will be reduced further.

I am surprised that there is not an increase in chromosome loss in the rad9 diploids if there are abortive CO/BIR events. Are events trapped leading to lethality?

It would be helpful to show the locations of the primers used for Rad51 ChIP on the donor chromosome in Fig 2a or 4a. Were fold enrichments for Rad51 normalized to the 0 hr time point as well as KCC4?

Please include the conditions for the D-loop extension PCR (template DNA concentration and number of cycles).

Fig 5a: The decreased efficiency of BIR in the mph1 mutant by the plating assay is surprising because it has previously been shown to increase BIR in several different assays.

Fig 6c: The percent events should be normalized to the repair efficiency. It would be better to combine the data in parts b and c to directly compare the efficiency of CO and NCO in the strains shown. The data in part e show that NCOs are not increased in the rad9 mutant, both classes of events appear to be reduced. There are no p values on the graph b; is the difference between rad9 and rad9 mph1 sgs1 significant?

Reviewer #3 (Remarks to the Author):

Ferrari et al.

I am pleased with the effort that the authors made to respond to my comments and those of the other reviewers. I have some minor comments to improve the manuscript.

1. The legend is missing for Figure 1B-the red boxes, white boxes and stipple boxes need to be specified.

2. The Southern blot in Figure 2B is extremely light and at least for this reviewer it's hard to be convinced of the BIR signal and how it was quantified. It would be worth showing all of the blots in the supplement to see how these data were obtained.

3. Figure 4 and 5 should indicate which primers were used to measure ChIP at the cut and donor sites. Figure 4A shows the different primers used for the D-loop extension analysis and cut site, but it was not made clear to this reviewer (after looking at the supplement, etc.) what primers were used for the donor site ChIP.

4. Lines 243 to 312 in the Discussion are essentially a rehash of the Results. This section could be shortened significantly to focus on the arguments presented from lines 313 to 353.

Response to Referees Letter

We are sending you a second revised version of the Manuscript No. NCOMMS-19-23713A, entitled '*Rad9/53BP1 promotes DNA repair via crossover recombination by limiting the Sgs1 and Mph1 helicases*'.

Once again, we thank the Reviewers for their fundamental contribution to revise our work.

Below is a detailed discussion of the various criticisms raised and of the modifications we made at the manuscript.

Kind Regards,

Achille Pelliccioli

Reviewers' comments in Blue, our response in Black:

Reviewer #1:

The authors have considered the three reviewers' comments very carefully and have performed additional experiments and have rewritten sections of the manuscript and added and modified figures. The response to the comments is good and the manuscript rewritten sections of the manuscript and added and modified figures. The response to the comments is good and the manuscript is much improved. It is clearer now which strains/constructs are used in which experiments, although there remains the unsatisfying finding of different rates/measurements with the same strain in different labs. It is known that media source is important, that the best source seems to be Difco and other formulations can have inhibitory effects on sporulation and cell viability. Nonetheless, the results stand in this manuscript and there are bound to be some difference between labs.

Importantly, the authors show that Rad9/53BP1 has an additional role in DSB repair and pathway choice through D-loop stabilization, targeting the Sgs1 and Mph1 DNA helices. This should have implications that extend to mammalian DSB repair pathways. A few minor comments are below.

1. line 89. I think ADE2/ade2 and ade2/ade2 should be italicized.

line 94. This is probably a difference between English and Italian verb tenses. All survivors are tested, not should be tested. The rationale for I-SceI re-induction is not stated although who do such experiments understand the reason.

3. line 178. "cells lacking RAD9". Here RAD is italicized, referring to the gene. However, I think in the case the focus is on the absence of the protein so it should be Rad9.

ANSWER: We did the changes to the text. Thank you.

Reviewer #2

The authors have addressed most of the issues raised at the first review. However, I still have some concerns, and suggestions for presentation of the data.

Figure 1c: It would be helpful to plot the percent of events adjusted for the cell viability, which is about 10% lower for *rad9* than wild type. I suspect that with this adjustment NCOs are not increased and BIR and CO products will be reduced further.

ANSWER: We did the suggested modification in Figure 1c. As correctly anticipated by the Reviewer, now the NCOs are almost similar in the wild type and *rad9* cells, while CO and BIR events in the *rad9* mutant are reduced further. We did the corresponding changes to the text.

I am surprised that there is not an increase in chromosome loss in the *rad9* diploids if there are abortive CO/BIR events. Are events trapped leading to lethality?

ANSWER: In this work we did not investigate molecularly the DSB repair events in the diploid cells. Therefore, we could not properly address the interesting question raised by the reviewer. In the revised discussion, we have added a short comment to acknowledge this possible scenario. Thank you.

It would be helpful to show the locations of the primers used for Rad51 ChIP on the donor chromosome in Fig 2a or 4a.

ANSWER: We added the position of primers P8 and P9 in Figure 4a. The corresponding sequences are listed in Supplementary Table 6.

Were fold enrichments for Rad51 normalized to the 0 hr time point as well as KCC4? Please include the conditions for the D-loop extension PCR (template DNA concentration and number of cycles).

ANSWER: The fold enrichments for Rad51 were not normalized to the 0 h. For clarity, in the revised figure legends we have indicated this information for Sgs1 and Mph1 ChIP analyses. The normalization procedure and data analysis are available as accompanying source data file. We also expanded the description of the protocol for D-loop analysis in the revised text.

Fig 5a: The decreased efficiency of BIR in the *mph1* mutant by the plating assay is surprising because it has previously been shown to increase BIR in several different assays.

ANSWER: Based on our statistical analysis, the viability of the *mph1* mutant in the BIR assay of panel 5a is not significantly reduced respect to the wild type cells (please, see corresponding source data file). However, the reviewer is right. We also expected moderately increased viability of the *mph1* cells in the BIR assay of panel 5a. As we discussed in the text, we could appreciate a slight contribution of the *mph1* mutation only when it is combined with the *sgs1* mutation, which always did the major effect in our experiments. Perhaps, the discrepancy between our results and the results obtained by others in the *mph1* mutant could be related to small variation in the experimental settings and cell growing conditions, as we also discussed in previous revision step for few other experiments.

Fig 6c: The percent events should be normalized to the repair efficiency. It would be better to combine the data in parts b and c to directly compare the efficiency of CO and NCO in the strains shown.

ANSWER: In the revised Figure 6, we did the requested modifications.

The data in part e show that NCOs are not increased in the *rad9* mutant, both classes of events appear to be reduced. There are no p values on the graph b; is the difference between *rad9* and *rad9 mph1 sgs1* significant?

ANSWER: Previous graphs 6b and 6c are now combined in single graph in the new panel 6b. We added the requested p value too.

Reviewer #3

I am pleased with the effort that the authors made to respond to my comments and those of the other reviewers. I have some minor comments to improve the manuscript.

1. The legend is missing for Figure 1B-the red boxes, white boxes and stipple boxes need to be specified.

ANSWER: We added the missing part to the legend. Thank you.

2. The Southern blot in Figure 2B is extremely light and at least for this reviewer it's hard to be convinced of the BIR signal and how it was quantified. It would be worth showing all of the blots in the supplement to see how these data were obtained.

ANSWER: Please, see the blots and the details for the signal quantification in related Source data file, as indicated in the Figure 2.

3. Figure 4 and 5 should indicate which primers were used to measure ChIP at the cut and donor sites. Figure 4A shows the different primers used for the D-loop extension analysis and cut site, but it was not made clear to this reviewer (after looking at the supplement, etc.) what primers were used for the donor site ChIP.

ANSWER: We added primers P8 and P9 in Figure 4a. The corresponding sequences are listed in Supplementary Table 6.

4. Lines 243 to 312 in the Discussion are essentially a rehash of the Results. This section could be shortened significantly to focus on the arguments presented from lines 313 to 353.

ANSWER: Accordingly, we shortened the indicated part of the Discussion.

REVIEWERS' COMMENTS:

Reviewer #2 (Remarks to the Author):

The authors have addressed all of my concerns. The only suggestions I have are to make some minor sentence modifications:

Line 64: suggested change: "...and then using specialized systems to physically monitor recombination products, we found..."

Lines 250 and 267: I don't think it is necessary to state the names of the investigators who developed the systems, having the references is sufficient.

Response to Referees Letter

We are sending you a third revised version of the Manuscript No. NCOMMS-19-23713B, entitled '*Rad9/53BP1 promotes DNA repair via crossover recombination by limiting the Sgs1 and Mph1 helicases*'.

Below is a detailed discussion of the various criticisms raised and of the modifications we made at the manuscript.

Kind Regards,

Achille Pelliccioli

Reviewers' comments in Blue, our response in Black:

Reviewer #2:

The authors have addressed all of my concerns. The only suggestions I have are to make some minor sentence modifications:

Line 64: suggested change: "...and then using specialized systems to physically monitor recombination products, we found..."

Lines 250 and 267: I don't think it is necessary to state the names of the investigators who developed the systems, having the references is sufficient.

ANSWER: done, thank you.